# Circulating Tumour DNA (ctDNA) as a Predictor of Clinical Outcome in Non-Small Cell Lung Cancer Undergoing Targeted Therapies: A Systematic Review and Meta-Analysis

**DOI:** 10.3390/cancers15092425

**Published:** 2023-04-23

**Authors:** Farzana Y. Zaman, Ashwin Subramaniam, Afsana Afroz, Zarka Samoon, Daniel Gough, Surein Arulananda, Muhammad Alamgeer

**Affiliations:** 1Department of Medical Oncology, Monash Health, Clayton 3168, Australia; 2School of Public Health and Preventive Medicine, Monash University, Clayton 3168, Australia; 3Department of Intensive Care, Peninsula Health, Frankston 3199, Australia; 4Peninsula Clinical School, Monash University, Frankston 3199, Australia; 5Centre for Cancer Research, Hudson Institute of Medical Research, Clayton 3168, Australia; 6Department of Molecular and Translational Science, Monash University, Clayton 3168, Australia; 7School of Clinical Sciences, Faculty of Medicine, Nursing and Health Sciences, Monash University, Clayton 3168, Australia

**Keywords:** non-small-cell lung cancer, NSCLC, liquid biopsy, circulating tumour DNA, ctDNA, progression-free survival, targeted therapies

## Abstract

**Simple Summary:**

Liquid biopsies have revolutionised the diagnostic and therapeutic landscape of non-small cell lung cancer (NSCLC), where several distinct genomic subtypes exist. Measuring such circulating biomarkers in serum/plasma is a feasible alternative to tissue biopsy, but the full clinical utility is yet to be established. This systematic review and meta-analysis aimed to evaluate the prognostic value of circulating tumour or cell-free DNA (ctDNA/cfDNA) in NSCLC. In a population of 3419 patients with molecularly altered and incurable advanced NSCLC. Negative ctDNA levels at baseline and early reduction after treatment correspond with clinical outcomes based on the results of our analysis. Though our results showed substantial heterogeneity, evolving data from specifically designed clinical trials may affirm these findings. As such, we suggest that future clinical trials should routinely incorporate ctDNA monitoring.

**Abstract:**

Background: Liquid biopsy (LB) analysis using (ctDNA)/cell-free DNA (cfDNA) is an emerging alternative to tissue profiling in (NSCLC). LB is used to guide treatment decisions, detect resistance mechanisms, and predicts responses, and, therefore, outcomes. This systematic review and meta-analysis evaluated the impact of LB quantification on clinical outcomes in molecularly altered advanced NSCLC undergoing targeted therapies. Methods: We searched Embase, MEDLINE, PubMed, and Cochrane Database, between 1 January 2020 and 31 August 2022. The primary outcome was progression-free survival (PFS). Secondary outcomes included overall survival (OS), objective response rate (ORR), sensitivity, and specificity. Age stratification was performed based on the mean age of the individual study population. The quality of studies was assessed using the Newcastle–Ottawa Scale (NOS). Results: A total of 27 studies (3419 patients) were included in the analysis. Association of baseline ctDNA with PFS was reported in 11 studies (1359 patients), while that of dynamic changes with PFS was reported in 16 studies (1659 patients). Baseline ctDNA-negative patients had a trend towards improved PFS (pooled hazard ratio [pHR] = 1.35; 95%CI: 0.83–1.87; *p* < 0.001; I^2^ = 96%) than ctDNA-positive patients. Early reduction/clearance of ctDNA levels after treatment was related to improved PFS (pHR = 2.71; 95%CI: 1.85–3.65; I^2^ = 89.4%) compared to those with no reduction/persistence in ctDNA levels. The sensitivity analysis based on study quality (NOS) demonstrated improved PFS only for good [pHR = 1.95; 95%CI: 1.52–2.38] and fair [pHR = 1.99; 95%CI: 1.09–2.89] quality studies, but not for poor quality studies. There was, however, a high level of heterogeneity (I^2^ = 89.4%) along with significant publication bias in our analysis. Conclusions: This large systematic review, despite heterogeneity, found that baseline negative ctDNA levels and early reduction in ctDNA following treatment could be strong prognostic markers for PFS and OS in patients undergoing targeted therapies for advanced NSCLC. Future randomised clinical trials should incorporate serial ctDNA monitoring to further establish the clinical utility in advanced NSCLC management.

## 1. Introduction

Globally, lung cancer is one of the most commonly diagnosed cancers (accounting for 11.6% of all new cancer diagnoses) and the leading cause of cancer-related mortality [1]. The vast majority (approximately 84%) of lung cancers are non-small-cell lung cancer (NSCLC). Advanced NSCLC is associated with a poor prognosis [2]. In recent years, precision medicine, driven by the identification of molecular targets, has shaped the paradigm of treating NSCLC. This has led to the advent of more effective systemic therapies. The identification of the epidermal growth factor receptor (*EGFR*) driver mutation in advanced NSCLCs has revolutionised treatment, with *EGFR* tyrosine kinase inhibitors (TKIs) being the standard first-line systemic therapy in this subgroup of patients [3]. With increasing molecular information, the number of molecular targets has also increased. Current guidelines recommend testing for *EGFR*, Kirsten rat sarcoma (*KRAS*) and v-raf murine sarcoma viral oncogene homolog B1 (*BRAF*) mutations, anaplastic lymphoma kinase (*ALK*) and c-ros oncogene 1 (*ROS-1*) fusions, mesenchymal–epithelial transition factor *(MET*) exon 14 skipping mutations, rearranged during transfection (*RET*) rearrangements, and programmed-death-ligand 1 (PD-L1) expression at time of diagnosis [4,5]. 

Traditionally, the identification of genomic aberrations to guide treatment has been reliant on tumour tissue biopsies. There are limitations of tumour-biopsy-based genotyping: risks and morbidity of sampling, feasibility due to anatomical location, logistic delays, and poor and low quantity of obtained tumour DNA. Moreover, tissue biopsy often provides limited molecular information due to inherent tumour spatial and temporal heterogeneity. Resistance to targeted treatments inevitably occurs through acquired resistance mechanisms, tumour heterogeneity, and changing molecular landscape [6]. Therefore, dynamic and serial sampling may aid in identifying these resistance mechanisms and thus lead to improved treatments and outcomes.

In contrast, liquid biopsies (LB) encompass methods of sampling and analysing isolates from biological fluid samples to molecularly profile tumours and ultimately guide clinical care [7]. Tumour cells, from both the primary and metastatic sites, shed various macromolecules into the bloodstream. These molecules can be isolated and tested via liquid biopsy techniques. The commonly used biological analytes include circulating tumour DNA (ctDNA), cell-free DNA (cfDNA), methylated DNA, and circulating exosomes [8]. LB, being minimally invasive and easily repeatable, is also pertinent when exploring dynamic resistance mechanisms. Indeed, LB techniques have the advantage of better capturing intra-patient spatial and temporal tumour heterogeneity more accurately than tissue sampling [9]. Emerging evidence supports the high specificity and clinical effectiveness of LB testing in the NSCLC sphere [10,11,12]. Several prospective clinical trials are establishing the role of LB as a practical alternative technique to tissue profiling in NSCLC [13,14,15].

Over time, several analytic methods have been developed for the clinical application of ctDNA. These methods range from narrow throughput quantitative and digital PCR assays to select mutation detection, with low cost and rapid turnaround time, to more expansive next-generation sequencing (NGS)-based techniques to test multiple alterations in several cancer-related genes, with variable costs and turnaround time according to the gene-panel selection [4]. All these techniques vary in sensitivities and specificities, depending on the gene-panel selection [4]. 

Sufficient evidence has evolved over the recent years for the clinical utility of genotyping advanced cancers using ctDNA assay in NSCLC. LB may be used as an alternative to tumour-biopsy strategies in certain situations of scarce tissue samples, as recommended by the College of American Pathologists (CAP)/International Association for the Study of Lung Cancer (IASLC)/Association for Molecular Pathology (AMP) guidelines for molecular testing [16,17]. Certainly, there are limitations of LB techniques: imperfect sensitivity and specificity and various assays which are not yet universally standardised [8].

Several applications of LB have been proposed based on numerous studies [13,14,15]. While the predominant use has been molecular profiling for therapeutic utility, detection of resistance mechanisms and measuring allele frequencies as well as minimal residual disease (MRD) detection after curative surgery [18]. However, it is not known whether the quantification of LB correlates with clinical outcomes. In this systematic review and meta-analysis, we aim to evaluate the relation of the levels of ctDNA, cfDNA, and methylated DNA as detected by LB techniques on clinical outcome, focusing on progression-free survival (PFS) of patients with advanced NSCLC, as well as studying their broader clinical utility and effectiveness. It is hypothesised that results obtained via LB predict survival outcomes.

## 2. Methodology

This study was conducted in adherence to the Preferred Reporting Items for Systematic Reviews and Meta-Analyses (PRISMA) Statement [19]. The protocol was registered on PROSPERO (registration number CRD42022347791). 

### 2.1. Eligibility Criteria

Studies eligible for analysis were defined using Population, Intervention, Comparison, and Outcome (PICO) strategy [20]. The study population was adult (>18 years of age) patients with histologically or cytologically confirmed advanced (incurable stage III or IV) NSCLC. Type of intervention was definitive LB detection method (i.e., the measurement of ctDNA, cfDNA, or methylated DNA in plasma or serum). Comparator was individual LB performance in this study. The primary outcome was progression-free survival (PFS). We included studies with direct information about hazard ratio (HR) and 95% confidence interval (95%CI) or with sufficient data, from which these numbers could be calculated. We included all randomised controlled trials (RCTs) and observational studies published between 1 January 2000 to 31 August 2022, in English. Single centre, retrospective studies, those with less than 50 patients, letters to the editor, commentaries, review articles, editorials, expert opinions, case series, case reports, meeting records of conference abstracts with insufficient information, non-human studies, unpublished, and non-peer-reviewed articles were all excluded. Studies lacking information about clinicopathological parameters were also excluded. 

### 2.2. Search Strategy and Sources

The search strategy was developed in collaboration with all authors. Two independent authors (F.Y.Z. and Z.S.) searched Embase, Ovid MEDLINE, PubMed, and Cochrane Database, using agreed-upon medical subject headings (MeSH) terms. These search terms and strategies and limits used are included in Appendix A. These terms were combined with the Boolean operator “OR”. The references of selected articles were also manually searched for any additional studies. References from selected articles were examined for further relevant articles. Any duplicates were removed. The search was finalised on 2 September 2022.

### 2.3. Study Selection and Data Extraction

The Rayyan QC (Rayyan Systems Inc, Cambridge, MA, USA) software [21] was employed to shortlist relevant and included articles and to filter the rationale for rejected studies. Two reviewers (F.Y.Z. and Z.S.) undertook data extraction of the included articles. Data were collected using a pre-organised data extraction form in Excel. Any conflicts were resolved via consensus between reviewers or adjudication by a fourth author (M.A.). In the case of overlapping patient data across two or more studies in our primary meta-analysis, we included the larger study. Corresponding authors were contacted for additional information in case data were incomplete. Data collection covered study characteristics (study design, study period, sample size, and country where the study was conducted), patient demographics (age, gender, cancer stage and histopathology, smoking status, Eastern Cooperative Oncology Group [ECOG] status, LB technique, and molecular alteration), and clinically relevant outcomes (progression-free survival [PFS], overall survival [OS], objective response rate [ORR], sensitivity, and specificity). These were independently extracted, tabulated, and verified by the two reviewers (F.Y.Z. and M.A.).

### 2.4. Quality Assessment and Risk of Bias in Individual Studies

The quality of studies was assessed using the Newcastle–Ottawa Scale (NOS) tool [22] by two independent reviewers (F.Y.Z. and M.A.) using the same set of decision rules. Any discrepancies were resolved by a third author (A.S.). Publication bias was examined using the symmetry of forest plots, Egger’s regression test, and leave-one-out forest plots [23]. To account for the heterogeneity, sensitivity analysis was performed based on study quality for all outcomes.

### 2.5. Study Aims and Outcomes

The primary aim was to examine the associations of ctDNA, cfDNA, and methylated DNA levels LB techniques at baseline and following treatment on clinically relevant outcomes. The primary outcome was PFS. Secondary outcomes included overall survival (OS), objective response rate (ORR), sensitivity, specificity, and resistance mechanism to treatment. Pre-determined subgroup analysis was based on age stratification (≤60 years vs. >60 years), the different platforms to detect ctDNA (NGS vs. non-NGS), and primary molecular alteration (*EGFR* vs. non-*EGFR* mutations) for PFS at baseline and following treatment.

### 2.6. Data Collection and Analysis

Most statistical analyses were performed using the statistical software package Stata-Version 17 (StataCorp, Texas, TX, USA). Mean (standard deviation [SD]) was used for numerical data and proportion for categorical data. Where the median (IQR) was reported, the mean (SD) was derived using an estimation formula [24]. Age stratification was performed based on the mean age of the individual study population. Due to high heterogeneity across the studies, a random effect model was used to calculate the estimated pooled HR and the event rates and to account for both within-study and between-study variances [25]. The HR and 95%CI in some studies were inverted to create uniformity in the reference and comparison groups [26]. The results were presented in Forest plots as pooled hazard ratio (pHR) for PFS and OS. Heterogeneity was tested using the χ^2^ test on Cochran’s Q statistic, which was calculated using H and I^2^ indices. The I^2^ index estimates the percentage of total variation across studies that were based on true between-study differences rather than on chance. Conventionally, I^2^ values of 0–25% indicate low heterogeneity, 26–75% indicate moderate heterogeneity, and 76–100% indicate substantial heterogeneity [27]. The publication bias was assessed using funnel plots and Egger’s regression test. The leave-one-out analysis was performed as a sensitivity analysis to demonstrate how each individual study affects the overall estimate of the rest of the studies. A further sensitivity analysis was performed based on study quality, as determined by NOS. A *p*-value < 0.05 was considered statistically significant.

## 3. Results

### 3.1. Characteristics of Included Studies

Of the 1906 references published until 2 September 2022, we assessed 116 full-text articles (Figure 1 PRISMA flowchart). Of those, 27 studies (32 cohorts) across 12 countries, which reported on 3419 patients with advanced (stage 4 or unresectable stage 3) NSCLC (99% had adenocarcinoma histological subtype), were included for qualitative and quantitative analysis (Table 1). Overall, identified records mainly included clinical trials with exploratory endpoints assessing the value of LB, prospective-observational studies, and case series. The basic characteristics of the studies are summarised in Table 1. Of these, 14 cohorts from 11 studies that reported on 1359 patients described the association of baseline ctDNA levels and clinical outcomes and were grouped into Category A (Cat A) studies [28,29,30,31,32,33,34,35,36]. The other 18 cohorts from 16 studies that reported on 1659 patients that described the association of dynamic changes in ctDNA with clinical outcomes were grouped as Category B (Cat B) studies [29,32,34,36,37,38,39,40,41,42,43,44,45,46,47,48]. To note, six studies [29,32,34,35,36,44] qualified for both categories and hence were included in both categories, respectively. Data from all 27 studies analysed PFS, 19 cohorts considered OS (5 in Cat A [29,32,35,36,44] and 14 in Cat B [29,34,35,36,38,39,40,42,43,44,45,48]. All studies included treatment with TKIs, with the majority (19 studies) involving *EGFR* [28,29,31,34,36,37,38,39,40,41,42,43,44,46]; 5 cohorts involved patients with *ALK* [30,32,37,49], 2 had *KRAS* [33,47], and 1 had c*MET* [35] alterations. The sample size per study ranged from 22 to 303 patients, and the studies were published between 2016 and 2022. A total of 6 cohorts reported only on the NGS LB technique [28,30,33,35,46,49], 18 reported on the non-NGS LB technique (PCR), and 2 studies reported on both NGS and PCR LB techniques [28,46]. Based on the NOS, thirteen cohorts were of good quality [29,30,31,34,35,36,39], four studies were of fair quality [32,37,40,41] and the remaining ten studies were of poor quality. Four studies used cfDNA [29,30,44,47], while the rest used ctDNA. (The term ‘ctDNA’ will be used from here onwards to denote both ctDNA and cfDNA).

### 3.2. Patient Demographics

More than half (55.8%) of the patients were female (reported in 21 studies). The cumulative mean [SD] age was 62.7 [36.2]. Almost a third of the patients (66.8%) (reported in 21 studies) never smoked. Additionally, 86% of patients (reported in 15 studies) had an ECOG of 0 or 1. The rest of the baseline characteristics of the included patients are summarised in Table 1.

### 3.3. The Association between ctDNA Detection and PFS

The association of baseline ctDNA levels (Cat A studies, 1626 patients) and early reduction/clearance of ctDNA levels (Cat B studies, 2070 patients) with PFS is illustrated in Figure 2. The raw pooled mean PFS for ctDNA positive was 11.3 months (range 5.6 to 24.1 months), and ctDNA negative was 15.4 months (range 5.0 to 36.1 months). The patients with positive ctDNA at baseline had a non-significant trend towards a shorter PFS compared to those where ctDNA was negative (pHR = 1.35; 95%CI: 0.83–1.87; *p* < 0.001; Figure 2a). Substantial heterogeneity (I^2^ = 96.0%) and publication bias (Egger’s test *p* < 0.0001; Appendix A) were observed. 

In Cat B studies, the median time from the start of treatment to the detection of ctDNA on LB was 8 weeks (range 4–16 weeks). The raw pooled mean PFS for ctDNA persistence was 6.9 (3.4–11.6) and 16.3 (range 8.0–294) for ctDNA reduction/clearance. The patients with persistent ctDNA had a shorter PFS compared to those with early reduction/clearance (pHR = 2.71; 95%CI: 1.85–3.56, *p* < 0.001; Figure 2b). There was, however, substantial heterogeneity (I^2^ = 89.4%) along with significant publication bias (Egger’s test *p* < 0.001, Appendix A).

The leave-one-out analysis demonstrating how each individual study affected the overall estimate of the rest of the studies is presented in Appendix A.

### 3.4. Subgroup Analysis 

Age stratification: Overall the age did not impact the outcome; however older patients aged > 60 years with positive ctDNA at baseline had a slight trend towards poor PFS compared to ctDNA negative patients [pHR = 1.62; 95%CI: 0.72–2.52, *p* < 0.001; I^2^ = 71.28%], while younger patients (age ≤60 years) showed no difference ([pHR = 1.11; 95%CI: 0.39–1.84, *p* < 0.001; I^2^ = 98.26%]; Appendix A). A similar trend was found in those with early reduction/clearance of ctDNA levels with PFS (age >60 years [pHR = 2.95; 95%CI: 2.13–3.77; I^2^ = 20 %] vs. age ≤60 years [pHR = 2.36; 95%CI: 0.95–3.78; I^2^ = 96%]; Appendix A).

ctDNA methodology (NGS vs. non-NGS): Patients (1202 patients) utilising the NGS platform were grouped separately from those utilising non-NGS platforms (PCR, ddPCR, Beaming). Studies using NGS (4 in Cat A and 5 in Cat B and non-NGS (7 in Cat A and 13 in Cat B were analysed. The impact on PFS was not significant on both platforms (NGS [pHR = 1.20; 95%CI: 0.51–1.89; I^2^ = 97.9%] vs. non-NGS [pHR = 1.59; 95%CI: 0.77–2.41; I^2^ = 66.3%], Appendix A) in Cat A. However, the PFS was significantly better for NGS [pHR = 2.59; 95%CI: 1.49–3.70, *p* < 0.001; I^2^ = 93.9%], and non-NGS methodology was better [pHR= 2.92; 95%CI: 1.74–4.09, *p* = 0.56; I^2^ = 0%]; Appendix A) in Cat B studies if there was early reduction/clearance of ctDNA.

Primary molecular alteration (*EGFR* vs. others): Patients from 19 studies with *EGFR* mutations (7 in Cat A and 15 in Cat B) were grouped separately from others (*ALK*, *KRAS*, and c*MET*; 3 in Cat A and 4 in Cat B). One study in Cat B was excluded due to mixed alterations and treatments [36]. The results showed inconsistencies according to molecular alteration. In studies involving *EGFR*, baseline ctDNA (Cat A) revealed no significant improvement in PFS [pHR = 1.26; 95%CI: 0.56–1.96, *p* < 0.001; I^2^ = 97.5%] while significantly better according to dynamic changes in ctDNA (Cat B) [pHR = 3.11; 95%CI 2.03–4.19; I^2^ = 66.6%]; however, the results were not significant in non-*EGFR* studies in either category. (Appendix A).

Study quality according to NOS: The study quality had a strong association with PFS in both Cat A and Cat B studies. Pooled analysis showed significant improvement in PFS in baseline ctDNA negative patients in good [pHR = 1.95; 95%CI:1.52–2.38] and fair [pHR = 1.99; 95%CI:1.09–2.89] quality studies. Similarly, early reduction/clearance in ctDNA also resulted in significant improvement in PFS in good [pHR = 2.71; 95%CI:1.86–3.56] and fair [pHR = 3.95; 95%CI:2.26–5.64] quality studies (Appendix A). There was no such trend in poor-quality studies. Moreover, there was no heterogeneity (I^2^ = 0.00) in good and fair quality studies, while it was quite high in poor quality studies I^2^ = 97.3%) (Appendix A).

The meta-regression analyses demonstrated no correlations of PFS with age, study quality, NGS status, or *EGFR* status for either Cat A or Cat B studies. These results are summarised in Table 2.

Secondary Outcomes: The secondary outcomes are summarised in Table 3.

Overall Survival: The association of baseline ctDNA levels (Cat A) with OS (5 studies, 577 patients) is illustrated in Figure 3. The raw pooled mean OS for ctDNA positive was 31.6 months (range 10.9 to 51.7 months) and 39.7 months (range 23.8 months to not-reached (NR)) for ctDNA negative. There was no statistically significant difference between the two groups with regard to OS (pHR 1.15; 95%CI: 0.85–1.45, *p* < 0.001; I^2^ = 18.4 Figure 3). There was no publication bias (Egger’s test *p* = 0.06; Appendix A). The association between early reduction/clearance of ctDNA levels (Cat B) with OS was analysed in 14 studies (n = 1417 patients). The patients with early reduction/clearance of ctDNA levels had improved OS (pHR = 1.72; 95%CI: 1.03–2.40, *p* < 0.001; Figure 2b). There was, however, substantial heterogeneity (I^2^ = 78.1%) along with significant publication bias (Egger’s test *p* < 0.001, Appendix A). The pooled HR showed a trend towards better OS only for good-quality studies (Appendix A) in the sensitivity analysis, while there was no association between poor- and fair-quality studies.

Other outcomes of relevance: A total of 4 studies (involving 907 patients) [29,33,34,48] reported ORR according to baseline ctDNA levels. The mean ORR in ctDNA negative cohort was 60.2% (range 30.0 to 88.7%), while it was 56.5% in the ctDNA positive cohort (range 28.0 to 86.6%). Additionally, 3 studies reported ORR according to dynamic changes in ctDNA [33,34,42], ORR was 63.7% (range 32.0 to 92.3%) in patients with early reduction in ctDNA, while it was 23.1% (range 5.0 to 47.0%) if the ctDNA was still high during treatment. The sensitivity and specificity of the testing platform were reported in four studies in Cat B [39,40,47]. All four studies used the PCR method for testing ctDNA. The mean sensitivity was 73.8% (range 69.0 to 81.3%), while the mean specificity was 99.5% (range 98.0 to 100%). Resistance mechanism to treatment was reported in 6 studies (9 cohorts) [33,37,39,40,41,45]. Detection of T790M in ctDNA after treatment with first/second generation *EGFR* TKIs was the commonest mechanism of resistance identified.

## 4. Discussion

In this systematic review and meta-analysis, we investigated the predictive and prognostic value of liquid biopsy (ctDNA and cfDNA) in the management of advanced NSCLC with known targetable mutations. We explored the reported association between the presence of ctDNA in the baseline blood samples and serial ctDNA measurements with clinical response to the targeted treatments and survival. Our study had these key findings. Firstly, baseline negative ctDNA levels were associated with a trend towards improved PFS but not OS. Secondly, early reduction of ctDNA was associated with improved PFS and OS. Finally, there was substantial heterogeneity and publication bias suggesting selection bias.

Several studies have reported the negative prognostic value of pre-treatment ctDNA across tumour types [50,51]. This may understandably reflect the aggressive biology, such as higher tumour burden [52], or other poor prognostic factors, such as tumour necrosis, lymphovascular invasion, and high proliferation index [53], compared to those with negative or undetectable ctDNA at baseline. Other contributing factors of interest have been intra-tumour genomic heterogeneity, plasticity, frequencies, and the evolution of genomic aberrations that often go undetected in conventional tissue sampling. Some studies also reported the co-occurrence of other resistant mutations, resulting in poor survival [32,54]. Such de novo resistant mechanisms are readily detectable in LB, reflecting the shedding of different ctDNA clones from different tumour regions. Earlier targeting of these de novo resistant mechanisms may result in improved clinical outcomes. Though adding cytotoxic chemotherapy to first-line gefitinib in non-selected *EGFR* mutant NSCLC resulted in improved survival [55], further predictive values of ctDNA with combination treatment will be quantified in ongoing trials [56]. In our analysis, there was a non-significant trend towards poor PFS based on baseline ctDNA-positive status. However, when controlled for study quality, there was a significant benefit in terms of improved PFS in baseline ctDNA-negative patients. This certainly indicates the limitations due to study design, implementation, or publication bias.

The longitudinal monitoring of ctDNA in the current analysis suggests a positive association between early (within 4–16 weeks) reduction in ctDNA with better prognosis. The findings provide evidence that patients with detectable ctDNA at baseline but undetectable during treatment have a better prognosis. Late reduction of ctDNA (e.g., >3 months after therapy) has not been investigated in many studies, except one, that reported no benefit [34]. A few studies also reported serial testing (>2 points during treatment) [29,34,41], with variable utility and outcomes. Consequently, the impact of early versus late clearance of ctDNA on overall prognosis needs to be further investigated in properly designed trials.

The increase in or re-emergence of ctDNA/mutations not only reflects treatment resistance but often precedes the radiological progression by several weeks [51,57]. Moreover, serial liquid biopsies could be more informative than the baseline snapshot to elucidate the genomic landscape as well as the evolutional trajectories under selective treatment pressure. Further understanding of quantitative changes in variant allele frequency (VAF) and the relative proportion of mutant vs. wild-type alleles throughout patient treatment may impact clinically meaningful outcomes. This will require additional refinement of the testing methodologies by utilising ultra-sensitive platforms and other enrichment techniques. The ongoing trials investigating the correlation of dynamic ctDNA response with tumour burden and radiological RECIST (response evaluation criteria in solid tumours) response will generate more interest. Hence, the clinical utility of longitudinal monitoring of ctDNA mutations (including the non-targetable) in the treatment landscape of NSCLC is not yet fully recognised.

Our analysis showed various analytical methods were employed to detect molecular alteration in ctDNA. These methods vary in their sensitivities and can be challenging to apply when the amount of ctDNA varies from patient to patient. In general, the PCR-based, targeted approaches, especially ddPCR are more sensitive [58]; NGS has the advantage of broader detection, e.g., tumour suppressor gene (TSG), copy number variation (CNV), translocations, amplifications, etc., and more utility in detecting resistant mechanisms. Sensitivity and specificity comparisons of various platforms has been reported elsewhere [57]. Cost-effectiveness and turn-around time are the other two areas to deal with in the current scenario.

So, what is the further role of ctDNA in the clinical setting besides its prognostic value? In the diagnostic setting, patients with limited tissue biopsy samples for molecular characterization may benefit from the examination of the ctDNA. This raises the question of whether the ctDNA could be used as a surrogate for tissue diagnosis. The predictive value of LB-detected alterations appears similar to the tissue in the literature. In the current analysis, the median PFS (mPFS) in tissue-positive and ctDNA-positive patients (*EGFR* only) ranging from 9.8 to 24.1 (mean 10.2 months) [29,31,34,36] equates to the mPFS reported in large phase 3 trials involving 1st/2nd generation *EGFR TKIs* (*OPTIMAL 13.1*, *EURTAC 9.7*, *IPASS 10.8*, *LUX-LUNG 3 11.2*, *LUX-LUNG 6 11.0*, *LUX LUNG-7 11.0*) [59,60,61,62,63,64]. Although the positive predictive value of LB is quite high, the high false-negative rates make it unsuitable for prime time yet. In the current analysis, the PCR-based assays had relatively lower LB negative rates (true negative) of a median of 32.3% (range 2.5 to 67.7%) compared to NGS-based assays that showed higher LB negative rates with a median of 49.5% (range 13.6 to 55.0%). Based on the current literature, the clinical use of LB is limited to situations where tissue biopsy is not feasible. IASLC has already put forward a statement on these scenarios [65]. In an ideal world, a platform that can detect a broader range of potentially targetable mutations, including *EGFR, ALK*, *ROS-1*, *RET*, *MET*, *HER2* (human epidermal growth factor receptor 2), *BRAF KRAS*, and *NTRK* (Neurotrophic tyrosine receptor kinase) would be desirable.

### 4.1. Strengths

Our analysis was quite comprehensive in terms of literature review to explore the emerging role of ctDNA in the management of advanced NSCLC. The main strengths of our analysis have been the robust methodology and large sample size to construct the objective measure of the quantitative evidence in the field. Our results provide valuable contributions to the existing literature that may assist in the application of LB on a broader scale in clinical practice. Moreover, our analysis also uncovers some of the gaps in the field, especially with regard to testing techniques and trial designs, which, if improved, could result in efficient utilization of the technology.

### 4.2. Limitations

The current analysis has a few limitations that need to be addressed. Firstly, there was a wide range of heterogeneity in the type of studies and the reported outcomes. The main reason for high heterogeneity was due to the publication bias, which could not be corrected despite further analysis. Various reasons can be hypothesized. Most included studies were observational cohorts with selected populations or trials that used ctDNA as an exploratory outcome or ad hoc subgroup outcome measure. This selection bias could have influenced the publication bias. Moreover, the Egger’s regression test, a commonly used quantitative method that tests for asymmetry in the funnel plot, has a limitation of identifying small study effects and may not tell directly if a publication bias exists [66]. Interestingly, the heterogeneity was lower for good- and fair-quality studies in the sensitivity analysis, emphasizing the need for sufficiently powered clinical trial designs to investigate LB as primary outcome measures. Secondly, the studies were heavily skewed towards those involving *EGFR* mutations. This perhaps could be explained by the limitation in the availability of testing techniques in non-*EGFR* alterations. Further inherent limitations of LB, such as short half-life and low concentration of ctDNA from extraction and inability to detect certain resistance mechanisms, including small cell transformation, need to be overcome before future application [67]. Evolving technologies with boosted sensitivities and deeper sequencing capabilities could lead us to expect broader inclusion in the coming years. Taken together, the results of the analysis should, hence, be interpreted with great caution.

## 5. Conclusions

This large systematic review, despite heterogeneity, found that baseline negative ctDNA levels and early reduction in ctDNA following treatment are strong prognostic markers for PFS and OS in patients undergoing targeted therapies for advanced NSCLC. Future randomised clinical trials should incorporate serial ctDNA monitoring to establish the clinical utility in advanced NSCLC management.

## Figures and Tables

**Figure 1 cancers-15-02425-f001:**
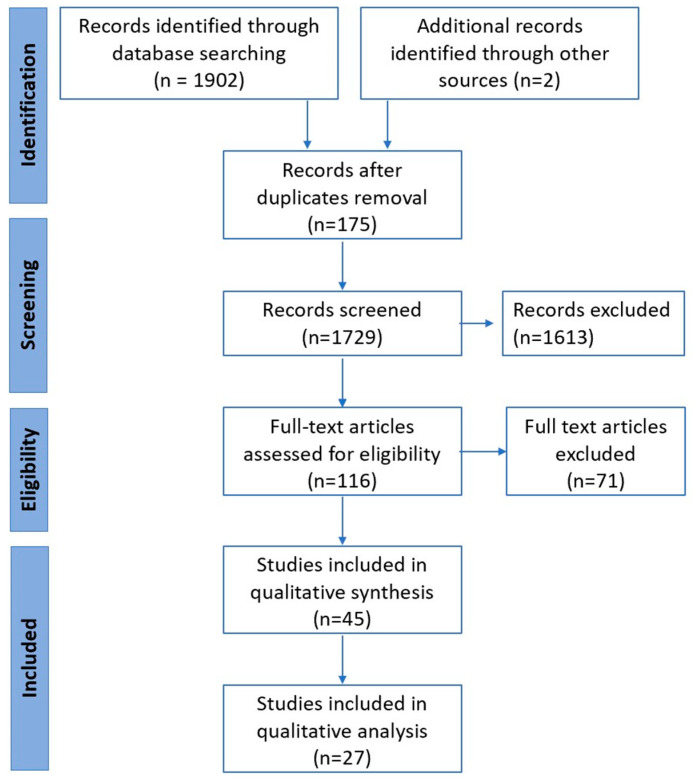
PRISMA flowchart of study inclusions and exclusions.

**Figure 2 cancers-15-02425-f002:**
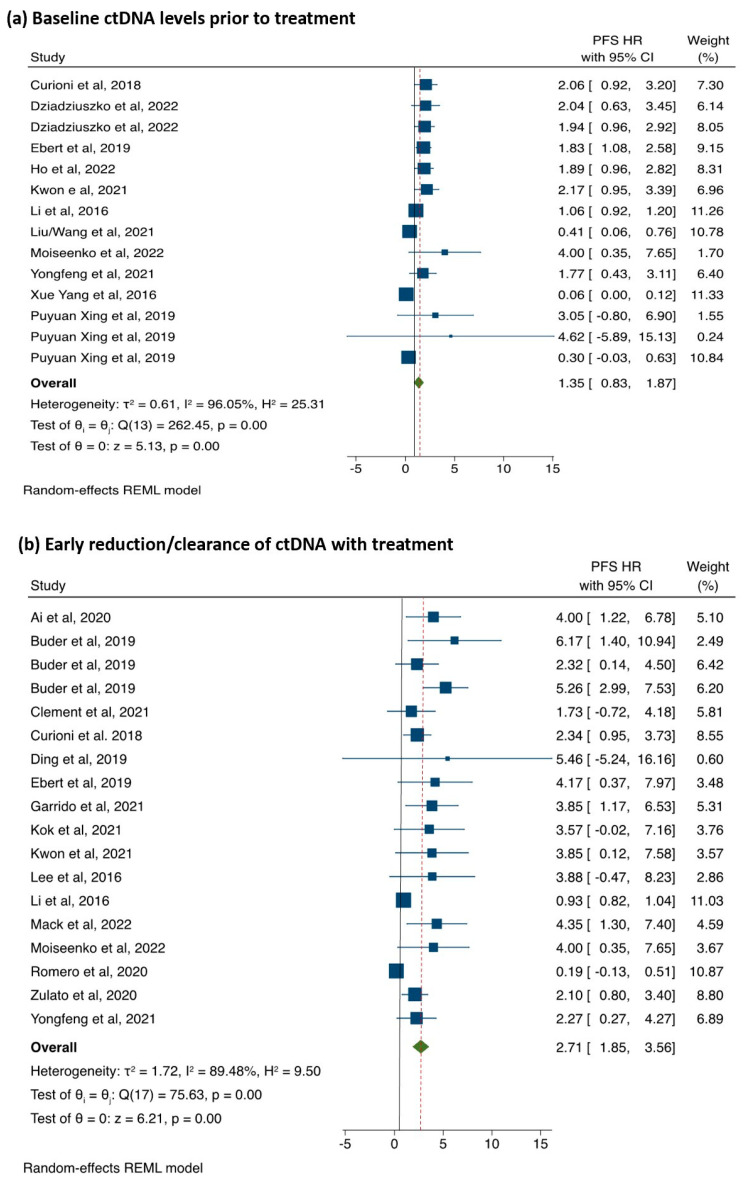
(**a**) The association between PFS and baseline ctDNA detection and (**b**) early reduction/clearance of ctDNA with treatment.

**Figure 3 cancers-15-02425-f003:**
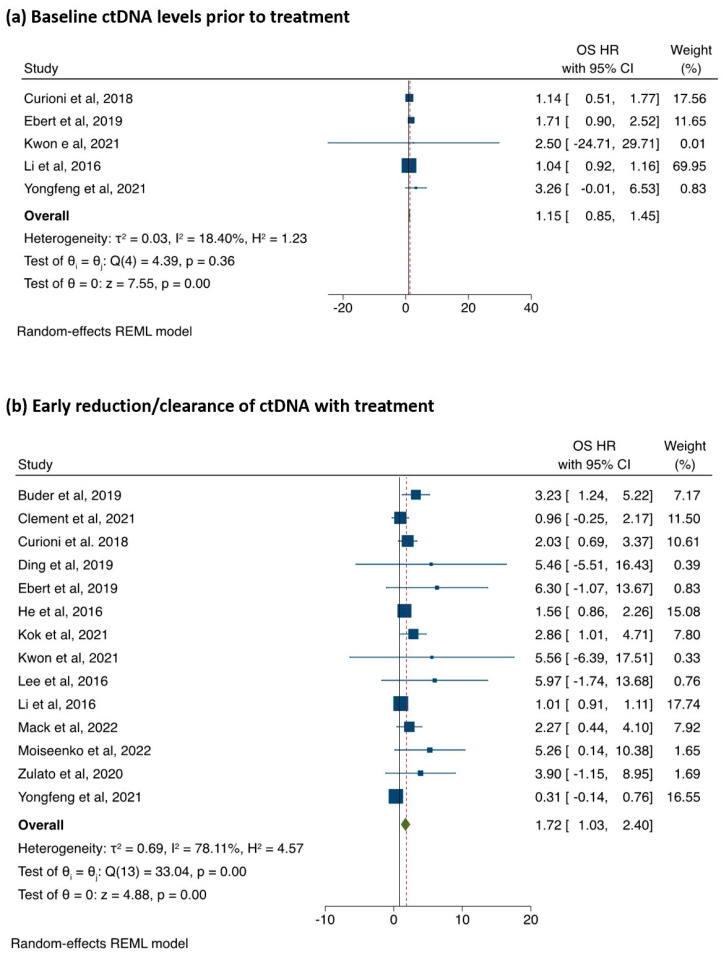
(**a**) The association between OS and baseline ctDNA detection (**b**) and reduction/clearance of ctDNA with treatment.

**Table 1 cancers-15-02425-t001:** Summary characteristics and descriptions for the included studies.

Study	Country	Patients	Molecular Alteration	LB Technique	Mean Age (SD) *	ECOG 0-1 Total (%)	Female’ Total (%)	Never Smoked (%)	Stage 4 Cancer (%)	Adenocarcinoma Histology (%)	NOS Grading
Category A Studies											
Curioni 2018 [29]	Switzerland	91	*EGFR*	PCR	65.4	87 (95.6)	61 (67.0)	60.3	-	97.8	Good
Dziadziuszko 2022 [30]	Poland	303	*ALK*	NGS	57.8 [44.4]	258 (85.1)	-	57.4	-	-	Good
Dziadziuszko 2022 [49]	Poland	303	*ALK*	NGS	57.8 [44.4]	258 (85.1)	-	57.4	-	-	Good
Ebert 2019 [36]	Denmark	225	*EGFR*	PCR	65.0	-	-	-	-	-	Good
Ho 2022 [31]	Taiwan	136	*EGFR*	PCR	-	97 (71.3)	76 (55.9)	72.8	99.3	100	Good
Kwon 2021 [32]	Korea	92	*ALK*	PCR	-	-	62 (67.4)	68.5	100	95.7	Fair
Li 2016 [44]	USA	103	*EGFR*	PCR	57.4	-	-	N/A	-	-	Poor
Liu 2021 [33]	China	135	*EGFR*, *KRAS, ALK*	NGS	-	42 (77.8)	29 (53.7)	79.6	-	-	Poor
Moiseenko 2022 [34]	Russia	99	*EGFR*	PCR	-	79 (79.8)	79 (79.8)	92.9	100	100	Good
Yongfeng Yu 2022 [35]	China	66	*METex14*	NGS	-	65 (98.5)	26 (39.4)	59.1	92.4	N/A	Good
Xue Yang 2016 [28]	China	73	*EGFR*	PCR/NGS	69.4	-	44 (60.3)	72.6	-	100	Poor
**Category B Studies**											
Ai 2020 [37]	China	300	*EGFR*, *ALK*	NGS	59.0 [37.0]	-	-	57	84	96.3	Fair
Buder 2019 [38]	Austria	106	*EGFR* T790M	PCR	60.3	-	-	-	-	-	Poor
Buder 2019 [38]	Austria	141	*EGFR*	PCR	64.7 [28.1]	-	80 (56.7)	-	-	-	Poor
Clement 2021 [39]	Denmark	76	*EGFR*	PCR	65.3 [28.1]	-	47 (61.8)	36.8	-	-	Good
Curioni 2018 [29]	Switzerland	91	*EGFR*	PCR	65.4	87 (95.6)	61 (67.0)	60.3	-	97.8	Good
Ding 2019 [40]	Australia	28	*EGFR*	PCR	67.0	22 (78.6)	16 (57.1)	75.0	100	100	Fair
Ebert 2019 [36]	Denmark	225	*EGFR*	PCR	65.0	-	-	-	-	-	Good
Garrido 2021 [41]	Spain	110	*EGFR*	PCR	65.5	102 (92.7)	79 (71.8)	61.8	-	-	Fair
He 2016 * [42]	China	200	*EGFR*	PCR	-	182 (91)	54 (27)	6.0	78	100	Poor
Kok 2021 [43]	Australia/ China	86	*EGFR*	PCR	-	86 (100)	49 (57.0)	72.1	97.7	93	Good
Kwon 2021 [32]	Korea	92	*ALK*	PCR	51.7 [43.0]	-	62 (67.4)	68.5	100	95.7	Fair
Lee 2016 [48]	Korea	81	*EGFR*	PCR	57.1 [36.3]	-	50 (61.7)	63.0	84	98.8	Poor
Li 2016 [44]	USA	103	*EGFR*	PCR	57.4	-	-	-	-	-	Poor
Mack 2022 [45]	USA	106	*EGFR*	PCR	64.3	96 (90.6)	69 (65.1)	-	100	93.4	Fair
Moiseenko 2022 [34]	Russia	99	*EGFR*	PCR	67.7	79 (79.8)	79 (79.8)	92.9	100	100	Good
Romero 2020 [46]	Spain	22	*EGFR* T790M	PCR/NGS	55.6	19 (86.4)	13 (59.1)	-	81.8	100	Poor
Zulato 2020 [47]	Italy	58	*KRAS*	PCR	67.3	54 (93.1)	27 (46.6)	63.8	-	-	Poor
Yongfeng 2021 [35]	China	66	*MET*ex14	NGS	69.4	65 (98.5)	26 (39.4)	59.1	92.4	-	Good

All abbreviations: LB—liquid biopsy, SD—standard deviation, NOS—Newcastle–Ottawa Scale, N/A—not applicable, ECOG—Eastern Cooperative Oncology Group, PCR—Polymerase Chain Reaction, NGS—next-generation sequencing, EGFR—epidermal growth factor receptor, ALK—anaplastic lymphoma kinase, KRAS—Kirsten rat sarcoma, MET– mesenchymal–epithelial transition. * Mean [SD] age was estimated from the median using the formula [23]. SD was estimated where the studies had reported interquartile range along with the median age.

**Table 2 cancers-15-02425-t002:** Univariable meta-regression analysis for progression-free survival treated as dependent variable accounting for age, study quality using Newcastle–Ottawa scale, NGS status, and *EGFR* status.

Covariate	Studies	Regression Coefficient (95%-CI)	*p*-Value
Category A Studies			
Age	10	−0.01 (−1.07 to 1.05)	0.98
Study quality	10	0.75 (0.27 to 1.25)	0.003
NGS status	10	0.17 (−0.81 to 1.14)	0.74
*EGFR* status	10	0.06 (−1.20 to 1.75)	0.72
**Category B studies**			
Age	17	0.98 (−1.12 to 3.08)	0.36
Study quality	17	−0.25 (−1.69 to 1.19)	0.73
NGS status	17	−0.81 (−1.70 to 3.32)	0.53
*EGFR* status	17	−1.26 (−3.29 to 0.77)	0.22

**Table 3 cancers-15-02425-t003:** Secondary outcomes. OS—overall survival, ORR—objective response rate, ctDNA—circulating tumour DNA, NR—not reached, N/A—not applicable.

Study	OS ctDNA Positive (Months)	OS ctDNA Negative (Months)	ORR ctDNA Positive (%)	ORR ctDNA Negative (%)	Sensitivity	Specificity	Resistance Mechanism
Category A Studies							
Curioni et al., 2018 [29]	27.0	36.6	-	-	-	-	-
Dziadziuszko et al., 2022 [30]	-	-	86.6	88.7	-	-	-
Dziadziuszko et al., 2022 [49]	-	-	72.3	80.3	-	-	-
Ebert et al., 2019 [36]	25.3	42.4	-	-	-	-	-
Ho et al., 2022 [31]	-	-	N/A	94.5	-	-	-
Kwon et al., 2021 [32]	39.5	NR	-	-	-	-	-
Moiseenko et al., 2022 [34]	51.7	56.2	28.0	67.0	-	-	T790M
Yongfeng et al., 2021 [35]	10.9	NR	52.2	30.0	-	-	-
Xue Yang et al., 2016 [28]	35.6	23.8	-	-	-	-	-
**Category B Studies**							
Buder et al., 2019 [38]	-	-	-	-	-	-	T790M
Buder et al., 2019 [38]	-	-	-	-	-	-	T790M
Buder et al., 2019 [38]	15.3	NR	-	-	-	-	T790M
Clement et al., 2021 [39]	30.2	30.5	-	-	-	-	-
Curioni et al., 2018 [29]	21.7	37.4	-	-	-	-	-
Ding et al., 2019 [40]	10.4	NR	-	-	69.0	100	T790M
Ebert et al., 2019 [36]	7.5	36.2	-	-			-
Garrido et al., 2021 [41]	-	-	-	-	70.9	98.0	T790M
He et al., 2016 [42]	27	34	-	-	-	-	T790M
Kok et al., 2021 [43]	15.8	30.1	5.0	32.0	-	-	-
Kwon et al., 2021 [32]	26.1	NR	-	-	-	-	-
Lee et al., 2016 [48]	11.2	23.7	-	-	74.1	100	-
Mack et al., 2022 [45]	15.9	32.6	-	-	-		-
Moiseenko et al., 2022 [34]	15.4	NR	28.0	67.0	-		T790M
Romero et al., 2020 [46]	-	-	-	-	-	-	T790M
Zulato et al., 2020 [47]	8.3	22.1	-	-	-	-	-
Yongfeng Yu et al., 2021 [35]	9.5	35.8	36.4	92.3	-	-	-

## Data Availability

The datasets are available from the corresponding author upon reasonable request.

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
