# Peer review of "Circulating Tumour DNA (ctDNA) as a Predictor of Clinical Outcome in Non-Small Cell Lung Cancer Undergoing Targeted Therapies: A Systematic Review and Meta-Analysis"

_cancers, 2023, doi:10.3390/cancers15092425_

Round 1

Reviewer 1 Report

The manuscript of Zaman et al "Circulating Tumour DNA (ctDNA) as a predictor of clinical 2 outcome in Non-Small Cell Lung Cancer Undergoing Targeted 3 Therapies: A Systematic Review and Meta-analysis" is a very well written and carefully performed review. There are just few typos, please check. I have no criticism

Author Response

Response: We thank the reviewer and appreciate these comments. All the typos have been addressed and corrected.

Reviewer 2 Report

Excellent review article, applying stringent study selection  conditions. Conclusions not clear in this field as yet.

 My only comment would be that Egger's test may be misleading for quantitative outcomes and this merits a sentence.

Author Response

Response: We appreciate the reviewer’s comments. We have drafted a sentence in the limitation section to address the issue you raised with regards to the Egger’s test (page 14), as below.

Moreover, the Egger’s regression test, a commonly used quantitative method that tests for asymmetry in the funnel plot, has a limitation of identifying small study effects and may not tell directly if a publication bias exists. Furthermore, the Egger’s does not account for the shape of the deviates, which is skewed in the presence of publication bias.”

(An extra reference [66] also included in the section).

Reviewer 3 Report

Question 1: Please explain according to the PICO criteria the inclusion criteria of patients in the study.

Question 2: did you perform sensitivity analysis to explore the influence of individual included studies on the overall pooled effect? Can you provide it? 

Question 3: A number of explanations have been highlighted in the text, directly in the pdf. Please provide a point-to-point answer.  

Author Response

Question 1: Please explain according to the PICO criteria the inclusion criteria of patients in the study.

Response: Inclusion criteria has been explained according to PICO, in the ‘Methodology’ section/eligibility criteria. As a result, an extra reference (no. 20) was added. Rest of the bibliography list has been adjusted accordingly.

Question 2: Did you perform sensitivity analysis to explore the influence of individual included studies on the overall pooled effect? Can you provide it? 

Response: Please note that the analysis we did considering NOS is a sensitivity analysis which can be found under sub-group analysis (Table 2). Furthermore, we have performed the LOO analysis that can also be considered as a sensitivity analysis. Results provided in the updated version as a supplementary figure 8.

Question 3: A number of explanations have been highlighted in the text, directly in the pdf. Please provide a point-to-point answer.  

Response: We thank the reviewer for picking some typographic errors. The other explanations/queries raised, have been answered in the updated version, some major ones explained below as well.

Lines 37-38: Sentence reworded for clarity.

Lines 176-177: Sentence reworded for clarity.

Line 217-218: under section Patient Demographics: Clarification provided for ‘21 studies’ mentioned.
